

# The first tetrapod from the mid-Miocene Clarkia *lagerstätte* (Idaho, USA)

Jonathan J. M. Calede[1,*], John D. Orcutt[2,*], Winifred A. Kehl[3] and Bill D. Richards[4]

[1] Department of Evolution, Ecology & Organismal Biology, Ohio State University, Marion, OH, USA
[2] Department of Biology, Gonzaga University, Spokane, WA, USA
[3] Unaffiliated, Grove City, OH, USA
[4] Department of Geology & Geography, North Idaho College, Coeur d'Alene, ID, USA
* These authors contributed equally to this work.

## ABSTRACT

The Clarkia *lagerstätte* (Latah Formation) of Idaho is well known for its beautifully preserved plant fossils as well as a fauna of insects and fish. Here we present the first known tetrapod fossil from these deposits. This specimen, recovered from the lower anoxic zone of the beds, is preserved as a carbonaceous film of a partial skeleton associated with a partial lower incisor and some tooth fragments. The morphology of the teeth indicates that the first tetrapod reported from Clarkia is a rodent. Its skeletal morphology as well as its bunodont and brachydont dentition suggests that it is a member of the squirrel family (Sciuridae). It is a large specimen that cannot be assigned to a known genus. Instead, it appears to represent the first occurrence of a new taxon with particularly gracile postcranial morphology likely indicative of an arboreal ecology. This new specimen is a rare glimpse into the poorly known arboreal mammal fossil record of the Neogene. It supports a greater taxonomic and ecological diversity of Miocene Sciuridae than previously recognized and offers new lines of inquiry in the paleoecological research enabled by the unique preservation conditions of the Clarkia biota.

## INTRODUCTION

The mid-Miocene is a critical interval for studies of the relationship between climate and paleoecological change. This is in large part due to the mid-Miocene climatic optimum (MMCO), a 2 °C warming event that peaked ca. 17 to 15 million years ago (Ma) and was the last sustained interval of climatic warming in the Cenozoic (*Zachos et al., 2001*; *Zachos, Dickens & Zeebe, 2008*). The magnitude of warming during the MMCO is consistent with predictions for climatic changes during the coming century (*IPCC, 2014*), and this has made the interval the subject of intensive paleontological and paleoecological study. The Inland Northwest of the United States (Idaho, eastern Oregon and Washington, and portions of surrounding states) is a natural laboratory for the study of the MMCO due not only to the detailed paleoclimatic (*Retallack, 2007, 2009*; *Takeuchi, Larson & Suzuki, 2007*; *Yang et al., 2011*) and paleoenvironmental (*Bestland et al., 2008*;

Corresponding author
John D. Orcutt, orcutt@gonzaga.edu

*Harris et al., 2017*; *Kohn & Fremd, 2007*; *Sheldon, 2006*) records available in the region but also to the wealth of terrestrial fossil floras and faunas preserved there. These include the vertebrate faunas of the Mascall (*Downs, 1956*; *Maguire, Samuels & Schmitz, 2018*), Sucker Creek (*Scharf, 1932*), and Virgin Valley formations (*Merriam, 1911*), which, along with other localities in the region, have served as the basis for several analyses of vertebrate macroecology (*Badgley & Finarelli, 2013*; *Calede, Hopkins & Davis, 2011*; *Harris, 2016*; *Maguire, 2015*; *Orcutt & Hopkins, 2013*).

While mammals have been the major focus of paleoecological research in the region, the Inland Northwest has also yielded an important insect fauna from the Latah Formation (*Carpenter, 1931*) and numerous fossil floras, notably from the Mascall (*Chaney, 1925*, *1959*; *Chaney & Axelrod, 1959*; *Knowlton, 1902*), Sucker Creek (*Arnold, 1937*; *Fields, 1996*), and Latah Formations (*Knowlton & Mann, 1925*). The abundance of contemporaneous fossils in the region from a wide variety of organisms could facilitate analyses of community structure through time across the MMCO. However, vertebrate, insect, and plant macrofossils rarely occur at the same sites, complicating such analyses. An exception to this rule is the Railroad Canyon section of Idaho, where data from phytoliths, stable isotopes from mammalian dental enamel, and vertebrate fossils have provided insight into ecosystem structure and community interactions before and during the MMCO (*Barnosky et al., 2007*; *Harris, 2016*; *Harris et al., 2017*). The Clarkia *lagerstätte* in the Idaho Panhandle (Fig. 1) represents another regional paleocommunity that could provide similar opportunities for integrative analyses of ecological change during the MMCO. This series of sites is best known for its rich flora (*Smiley & Rember, 1985a*). Leaves and needles are often preserved with original organic material intact, providing the unique opportunity to conduct biochemical analyses of Miocene plant tissue (*Huang et al., 1995*; *Kim et al., 2004*; *Lockheart, Van Bergen & Evershed, 2000*; *Logan, Boon & Eglinton, 1993*; *Logan, Smiley & Eglinton, 1995*; *Soltis, Soltis & Smiley, 1992*; *Yang & Huang, 2003*). Other terrestrial organisms recovered from the *lagerstätte* include fungus macrofossils (*Williams, 1985*), floral and fungal microfossils (*Gray, 1985*), insect fossils, and ichnofossils (*Lewis, 1985*). Although providing unparalleled opportunities for paleoecological analyses of a mid-Miocene forest ecosystem (*Batten, Gray & Harland, 1999*; *Smiley & Rember, 1985b*; *Smith & Elder, 1985*), the Clarkia fossil record has until now remained incomplete in one notable way. While salmonid, cyprinid, and centrarchid fish have been reported from the *lagerstätte* (*Smith & Miller, 1985*), no tetrapods have previously been reported from any of the Clarkia localities. Here we fill in this taxonomic gap by presenting the first known occurrence of a mammal, and indeed the first known tetrapod of any kind, from Clarkia.

## GEOLOGICAL SETTING

The Clarkia *lagerstätte* consists of several outcrops of the Latah Formation near the town of Clarkia in the Idaho Panhandle (*Smiley & Rember, 1985a*; Fig. 1). The portion of the Latah Formation that crops out at Clarkia was deposited on a lake bed formed when a Columbia River Basalt flow formed a natural dam of the proto-St. Maries

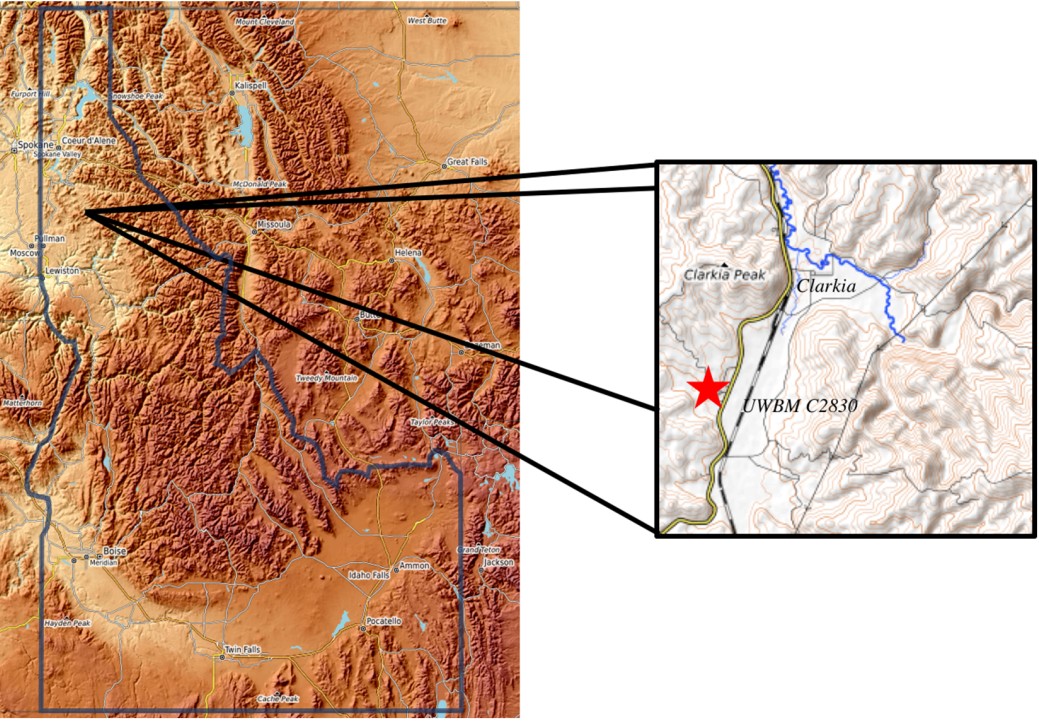

**Figure 1** **Location of UWBM C2830.** UWBM C2830, the Clarkia type locality, is equivalent to UIMM P-33. The section from which the Clarkia rodent was recovered is at the north end of the Kienbaum family racetrack in Clearwater County, Idaho.     

River near the present town of St. Maries, Idaho (*Yang et al., 1995*). Sediments at Clarkia consist predominately of clay and silt (much of it derived from mica of the underlying Precambrian schist), with some sandy layers at the base of the section (*Smiley, Gray & Huggins, 1975*). Several ash layers are present throughout the section and can be attributed primarily to eruptions associated with the Yellowstone hotspot and to a lesser extent with the Cascade volcanoes (*Ladderud et al., 2015*). *Smiley, Gray & Huggins (1975)* divided the Clarkia *lagerstätte* into two zones: a lower unoxidized zone and an upper oxidized zone. While fossils are found throughout the section, the exquisitely detailed flora with preserved biomolecules for which Clarkia is known are found only in the lower zone.

  The fossil described herein was found at the Clarkia type locality (UWBM C2830, originally described as UIMM P-33) at the Kienbaum family racetrack during a North Idaho College trip to the locality in 2009. It was uncovered within Unit 2 of *Smiley & Rember (1985a)* in the lower part of the lower unoxidized zone (Fig. 2). The fossil was found immediately below an ash layer correlated with an ash from the Bully Creek Formation, dated to 15.65 ± 0.07 Ma. The basalt dam that created the lake is estimated to have been emplaced ~16.0 Ma, though the precise basalt flow responsible has not been identified and dated. (*Ladderud et al., 2015*; *Nash & Perkins, 2012*). These two dates indicate that the first known tetrapod from Clarkia likely dates to the Early Barstovian North American Land Mammal Age (NALMA). The boundary between the Barstovian and preceding Hemingfordian NALMA lies at 16 Ma (*Tedford et al., 2004*), making a

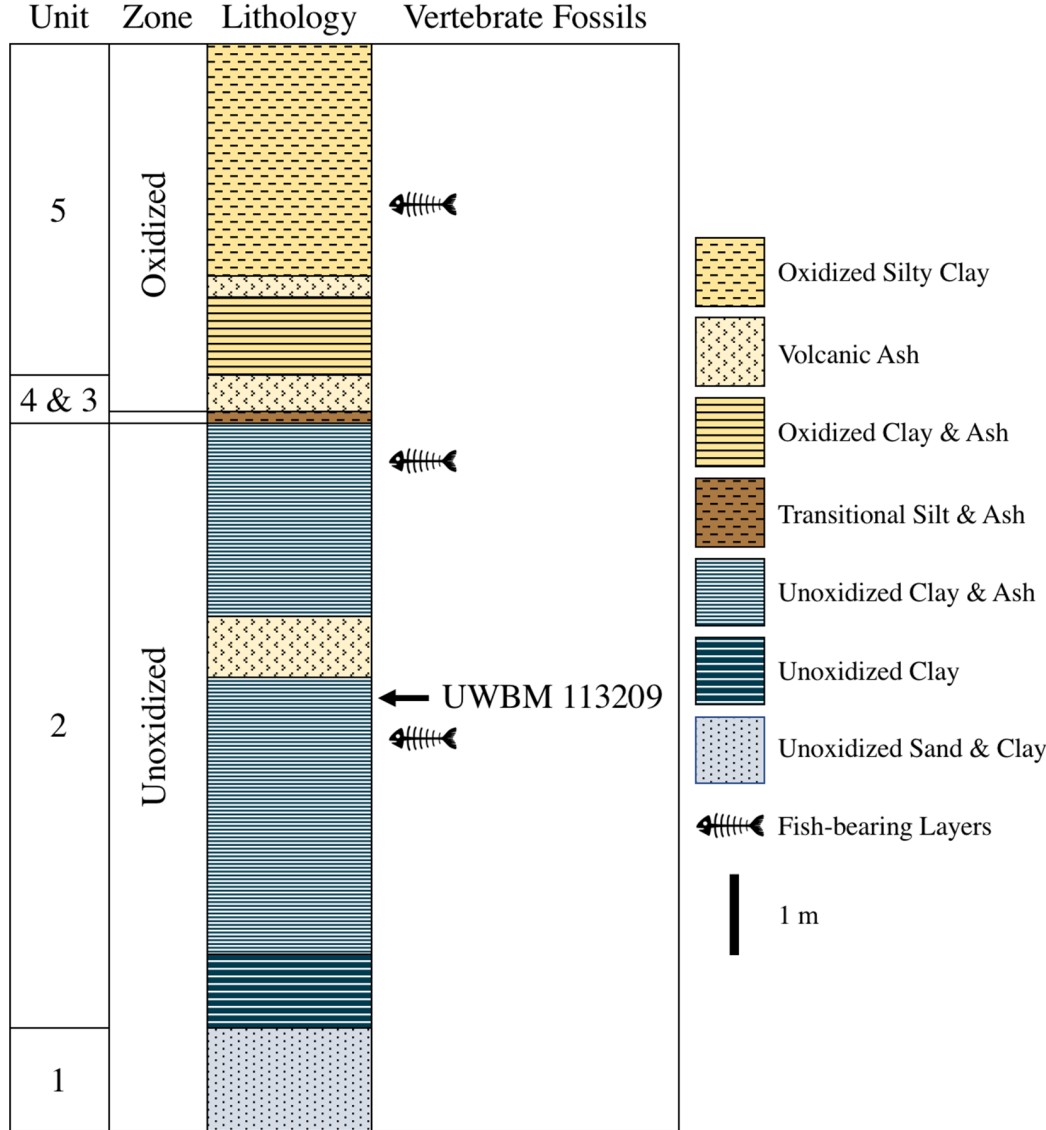

**Figure 2 Stratigraphy of the Latah Formation exposed at UWBM C2830.** Numbering, lithology, and color of units follows *Smiley, Gray & Huggins (1975)* and *Smiley & Rember (1985a)*. The total thickness of the section exposed at the Kienbaum racetrack is roughly 9 m.

Late Hemingfordian age possible as well. However, the specimen was recovered well above the base of the Clarkia sequence and as such, a Barstovian age is considerably more likely.

## MATERIALS AND METHODS

The taxonomic frameworks for the rodents from the Inland Northwest discussed in this paper come from *Flynn & Jacobs (2008a, 2008b)*, *Flynn, Lindsay & Martin (2008)*, *Goodwin (2008)*, and *Hopkins (2008a)*. We collected the measurements given in this paper either from the literature or directly from specimens using Mitutoyo Digimatic CD-4 CX calipers. The specimen is reposited at the University of Washington Burke Museum of Natural History and Culture (UWBM) in Seattle, Washington (USA).

# SYSTEMATIC PALEONTOLOGY

Class MAMMALIA *Linnaeus, 1758*

Order RODENTIA *Bowdich, 1821*

Family Sciuridae *de Waldheim, 1817*

(Fig. 3)

**Material**—From the Latah Formation, Idaho: UWBM C2830: UWBM 113209 (partial skeleton preserved as a carbonaceous film including skull, partial dentary, partial left and right scapulae, humeri, radii, and ulnae; left(?) manus, and partial vertebral column associated with ribs and sternum; fragments of the cheek teeth and a partial lower incisor are preserved as three-dimensional elements).

   **Description**—UWBM 113209 is preserved as a carbonaceous film of the skeleton. Only a partial lower incisor and fragments of the cheek teeth are preserved as three-dimensional elements. A mold of the upper incisor is also preserved. The chisel-shaped ever-growing lower incisor indicates that the first known tetrapod from the Clarkia *lagerstätte* is a rodent (*Luckett & Hartenberger, 1993*; *Landry, 1999*). The incisor is 3.3 mm thick dorsoventrally. Its anterior surface is smooth and convex. The diameter of the semi-circle formed by the incisor is 26.3 mm. The fragments of cheek teeth are scattered and no complete tooth is fully preserved. A few cuspules are preserved. They indicate that the specimen has a bunodont tooth shape associated with a brachydont crown height. We could not determine with certainty whether the fragments belong to the upper or lower cheek teeth. The skull is large (Table 1). No bone suture or process can be identified. The bones posterior to the upper diastema are too poorly preserved to be identified as specific elements. The dorsal surface of the cranium is somewhat convex. We estimate the upper diastema to be 20.2 mm long; the poor preservation of the cheek teeth prevents a more accurate measurement. The posterior portion of the skull and its articulation with the vertebral column are too poorly preserved to be described. Both the left and right forelimbs are partially preserved as carbonaceous films including the scapulae, humeri, radii, and ulnae. The posterior border of the scapula is convex. The nature of the preservation as a carbonaceous film prevents a detailed description of the morphology, processes, and articulation of the bones of the forelimb. Although the vertebrae are preserved as massive film without visible processes, several ribs can be individualized and the sternum is well preserved.

   **Comparison**—Although most of the detailed morphology of the skeleton of UWBM 113209 cannot be described and the teeth are fragmented, the size of the specimen as well as elements of its dentition allow the taxonomic identity of UWBM 113209 to be constrained. Based on skull length, we estimate the body mass of the animal at around 492 g (using the regression formula provided by *Bertrand, Schillaci & Silcox, 2015*). Only a few rodent families present in North America during the late Hemingfordian and Barstovian reach such a large body size including the Aplodontiidae (including Mylagaulidae), Castoridae, and Sciuridae. Although some rodents of the clade Geomorpha (Geomyidae, Heteromyidae, and their fossil relatives; *Flynn, Lindsay & Martin, 2008*) do reach such

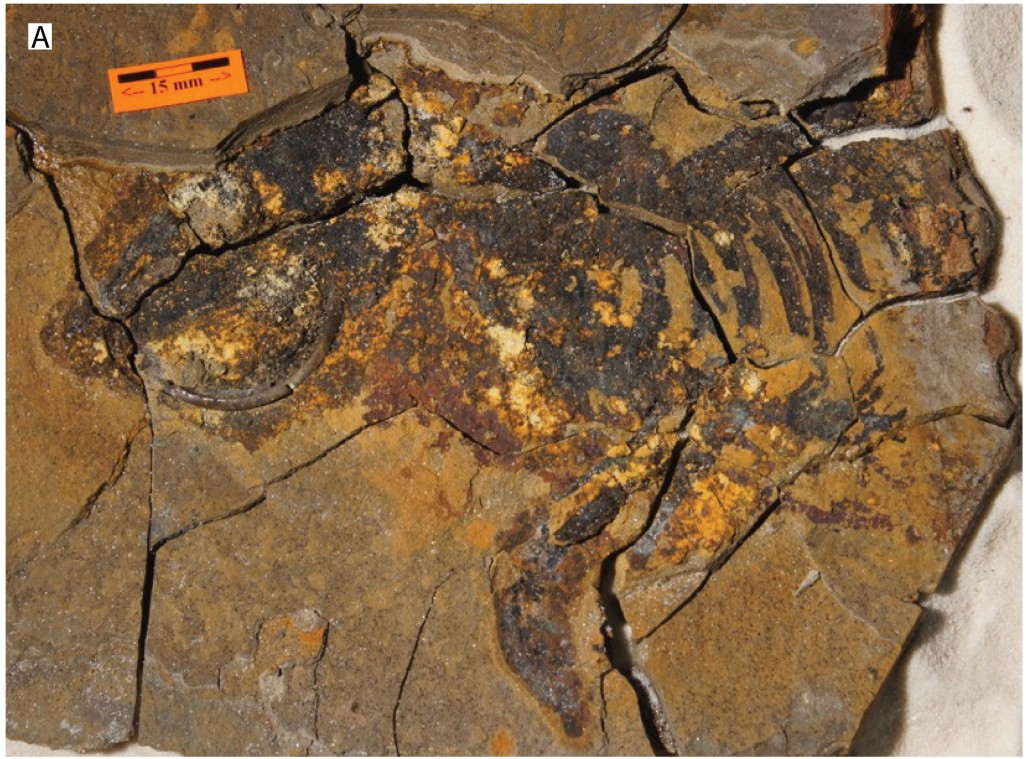

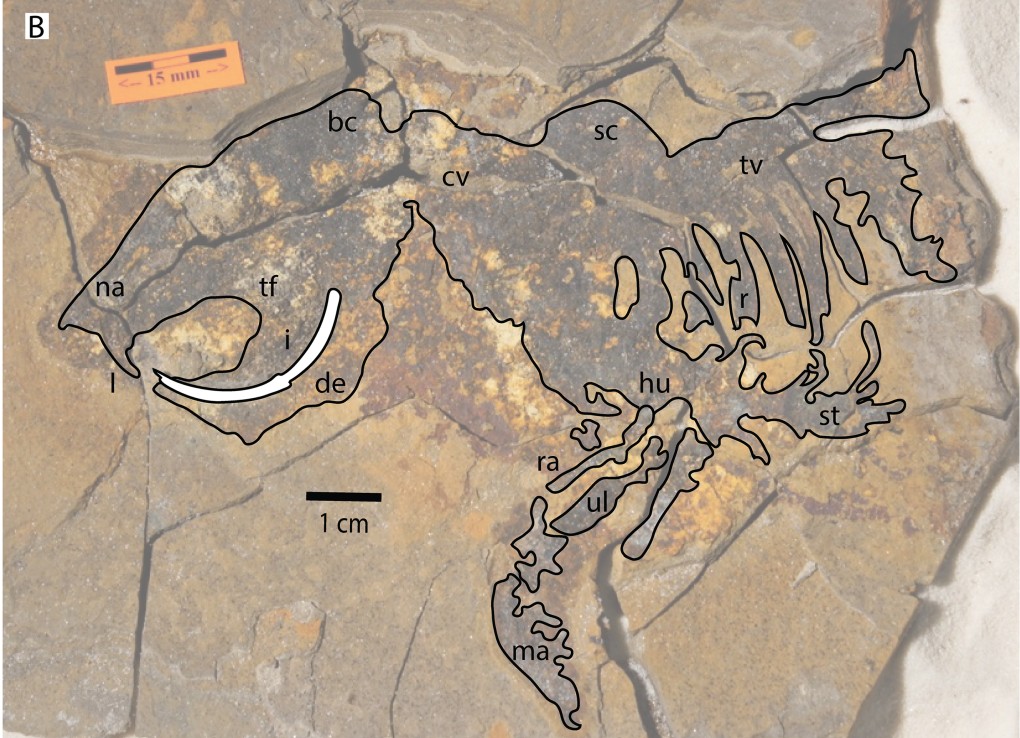

**Figure 3** **Photograph and line drawing of UWBM 113209.** (A) Photo of UWBM 113209, (B) line drawing showing approximate outline of the skeleton. Abbreviations: bc, back of the cranium; cv, cervical vertebrae; de, dentary; hu, humerus; I, upper incisor; i, lower incisor; ma, manus; na, nasals; ra, radius; sc, scapula; st, sternum; tf, tooth fragments; tv, thoracic vertebrae; ul, ulna. Photo by Bill Richards and illustration by Winifred Kehl.               

**Table 1 Summary of the measurements of UWBM 113209.**

| Measurement | Value |
|---|---|
| Dorsoventral thickness of lower incisor | 3.3 |
| Diameter of semi-circle formed by incisor | 26.3 |
| Skull length | 56.5 |
| Upper diastema | 20.2* |
| Lower tooth row length | 13.4* |
| Body mass | 492 |

Notes:
  Linear measurements in millimeter. Mass in gram.
  * Denotes estimate.

large sizes today, none of the taxa present during the Barstovian in the Inland Northwest, or even North America as a whole, did (*Barnosky et al., 2007*; *Calede, Hopkins & Davis, 2011*; *Munthe, 1977*). The largest geomorph from the Barstovian, *Geomys* (*Nerterogeomys*) cf. *G.* (*N.*) *paenebursarius* (*Tedford, 1981*) is estimated to have been two-thirds the size of the modern *Geomys bursarius* (*Strain, 1966*), an animal that weighs as much as 473 g (*Connior, 2011*), putting the maximum size of *Geomys* (*Nerterogeomys*) cf. *G.* (*N.*) *paenebursarius* at around 312 g. *Geomys* (*Nerterogeomys*) cf. *G.* (*N.*) *paenebursarius* further differs from UWBM 113209 by the shape of its lower incisor. *Geomys* (*Nerterogeomys*) has a flattened lower incisor (*Dalquest, 1978*; *Flynn, Lindsay & Martin, 2008*) whereas the Clarkia rodent has a convex incisor.

We used the body mass of UWBM 113209 calculated using skull length to estimate the length of the lower tooth row (LTRL; using the regression formula of *Hopkins, 2008b*, Table 2 for non-muroid rodents under 500 g). Although this estimate may not be very accurate because it is derived from an estimate of the body mass of UWBM 113209, which is itself determined from a measurement of the skull length based on a carbonaceous impression, the absence of a preserved tooth row prevents a direct measurement on the specimen. The lower tooth row length estimate (Table 1) allows comparisons of the body mass of UWBM 113209 with the database of *Hopkins (2007)* who surveyed the body mass of North American Aplodontiidae, Castoridae, and Sciuridae from the late Eocene through the end of the Miocene.

Only two aplodontiid genera (excluding mylagaulids) are known from the Barstovian (*Flynn & Jacobs, 2008a*; *Hopkins, 2008a*): *Liodontia* and *Tardontia*. Both are known from the Inland Northwest (*Calede, Hopkins & Davis, 2011*; *Shotwell, 1958*) and overlap in size with UWBM 113209 (*Calede, Hopkins & Davis, 2011*; *Hopkins, 2007*). However, they differ greatly in morphology. The diastemata of *Liodontia* and *Tardontia* are much shorter than that of UWBM 113209 (*Gazin, 1932*; *Morea, 1981*; *Shotwell, 1958*). The dentaries of both *Liodontia* and *Tardontia* are also overall much more robust than that of UWBM 113209 (*Gazin, 1932*; *Shotwell, 1958*). Additionally, both *Liodontia* and *Tardontia* display typical derived aplodontiid hypsodont dentitions (*Flynn & Jacobs, 2008a*; *Gazin, 1932*; *Hopkins, 2008a*; *Shotwell, 1958*) that are quite unlike the bunodont–brachydont dentition of UWBM 113209.

Although a diverse fauna of mylagaulids is known from the Inland Northwest (*Barnosky et al., 2007*; *Calede & Hopkins, 2012a*; *Calede, Hopkins & Davis, 2011*), most genera are much larger than UWBM 113209 (*Calede, Hopkins & Davis, 2011*). Only the genus *Mesogaulus* is similar in size to UWBM 113209 (*Dorr, 1956*; *Hopkins, 2007*). Despite similar sizes, the morphologies of the two rodents are quite different. Thus, the diastema of *Mesogaulus* is much shorter (*Dorr, 1956*), its skull and dentary are more robust than those of UWBM 113209 (*Galbreath, 1984*), and its lower incisor is larger (*Wilson, 1960*). The teeth of mesogauline mylagaulids (including *Mesogaulus*) are also very much unlike those of UWBM 113209. They are typically very large, high-crowned, and display a complex occlusal surface composed of enamel lakes (*Calede & Hopkins, 2012a*; *Shotwell, 1958*). The postcranial skeleton of UWBM 113209 provides additional evidence that the Clarkia rodent is not a mylagaulid, given its gracile forelimbs unlike those found in mylagaulids (*Calede & Hopkins, 2012b*; *Fagan, 1960*; *Galbreath, 1984*; *Korth, 2000*).

Among the genera of the family Castoridae known from the early Barstovian of North America (*Flynn & Jacobs, 2008b*), *Anchitheriomys* is much larger than UWBM 113209 and only *Euroxenomys* overlaps in size with UWBM 113209 (*Hopkins, 2007*). This genus is known from other Barstovian-aged deposits of the northwest (*Maguire, Samuels & Schmitz, 2018*). However, similarly to *Mesogaulus*, it is much too robust to compare well with UWBM 113209 (*Prieto, Casanovas-Vilar & Gross, 2014*). Although several species of *Monosaulax*, another castorid genus, are known from the early Barstovian, including from the Northwest (*Shotwell, 1968*), they are much larger than UWBM 113209 (*Hopkins, 2007*). Some species of *Monosaulax* from the late Barstovian are smaller than UWBM 113209 (*Hopkins, 2007*). Nonetheless, similarly to *Euroxenomys*, the morphology of the dentary and skull of *Monosaulax* is much more robust than UWBM 113209; the diastema of *Monosaulax* is also shorter than that of UWBM 113209 (*Korth, 2002*; *Matthew & Cook, 1909*; *Shotwell, 1968*; *Stirton, 1935*). The bunodont–brachydont tooth morphology of UWBM 113209 is quite unlike the lophodont dentition characteristic of Castoridae, providing one more line of evidence that the Clarkia rodent is not a beaver.

The low crowned and bunodont teeth of UWBM 113209 combined with its large size and gracile build support the interpretation that the first known tetrapod remains from the Clarkia *lagerstätte* belong to the family Sciuridae. The early Barstovian-aged sciurid fauna of the Inland Northwest is very species-rich (*Biedron, 2016*; *Goodwin, 2008*; *Maguire, Samuels & Schmitz, 2018*; *Orcutt & Hopkins, 2013*). Nonetheless, only a few taxa from this or neighboring regions are similar in size to UWBM 113209. *Palaeoarctomys montanus* (*Douglass, 1903*) from the Barstovian of Montana is slightly larger (LTRL = 15.1 mm) than UWBM 113209 (*Hopkins, 2007*). Another taxon from Montana, *Arctomyoides arctomyoides* (*Douglass, 1903*), is also slightly larger than UWBM 113209 (LTRL = 15.4 mm, *Bryant, 1945*). Within the genus *Protospermophilus*, *Protospermophilus oregonensis* (originally described as *A. oregonensis*, *Downs, 1956*) is slightly smaller than the Clarkia rodent (LTRL = 12.2 mm, *Hopkins, 2007*; LTRL = 11.9 mm, *Downs, 1956*). A set of sciurid specimens from the Arikareean of Nebraska named "*P. vondrai*" in an unpublished dissertation (*Martin, 1973*) is also only slightly smaller than the Clarkia

rodent (LTRL = 12.94 mm, *Hopkins, 2007*). Although referred to *Protospermophilus* by *Martin (1973)*, this set of specimens was later assigned to *Cedromus* by *Hayes (2005)*. Only one other Miocene-aged sciurid genus overlaps in size with the Clarkia rodent: *Protosciurus* (*Hopkins, 2007*; *Korth & Samuels, 2015*). *Protosciurus* is present in the northwest of the United States during the Arikareean and Hemingfordian (*Black, 1963*; *Goodwin, 2008*; *Korth & Samuels, 2015*); it is also reported in the early Barstovian of Texas (*Goodwin, 2008*). *Spermophilus*-grade ground squirrels (i.e., sciurid species formerly referred to the genus *Spermophilus* prior to reassessment of ground squirrel phylogeny by *Helgen et al., 2009*) are common during the Barstovian (*Orcutt & Hopkins, 2013*), but they are all smaller than UWBM 113209 (*Hopkins, 2007*).

Although *Palaeoarctomys montanus* is similar in tooth row length to the Clarkia rodent, its skull, measuring 100 mm (*Douglass, 1903*), is much longer than that of the Clarkia rodent. Additionally, its diastema (16.8 mm, *Black, 1963*) is shorter, if only slightly, than that of UWBM 113209. *Palaeoarctomys montanus* also differs from UWBM 113209 in its deep and robust dentary (*Goodwin, 2008*). This dentary houses a very large incisor (depth 7.3 mm, *Downs, 1956*), much larger than that of UWBM 113209. The lower incisor of *Palaeoarctomys montanus* bears striations on its anterior surface (*Goodwin, 2008*) whereas the lower incisor of the Clarkia rodent is smooth. Taken together, these features suggest that *Palaeoarctomys* is an unlikely candidate for the taxonomic affinities of the Clarkia sciurid.

*Arctomyoides arctomyoides* is larger than UWBM 113209 but its diastema is much shorter than that of UWBM 113209 (13.0 mm; *Black, 1963*). The depth of the lower incisor of *A. arctomyoides* (3.8 mm; *Black, 1963*) is greater than that of UWBM 113209. UWBM 113209 displays a long and shallow diastemal depression (Fig. 2), a characteristic of *Arctomyoides* (*Bryant, 1945*; *Goodwin, 2008*) and lacks a medial groove on the lower incisor alike *Arctomyoides* (*Goodwin, 2008*). However, alike *Palaeoarctomys montanus*, the lower incisor of *A. arctomyoides* is finely striated whereas that of UWBM 113209 is smooth. The poor preservation of UWBM 113209 prevents a rigorous comparison with the diagnostic characters of *Arctomyoides* summarized by *Goodwin (2008)*, especially with regard to the cheek teeth, but the smooth incisor of the Clarkia sciurid suggests that it is not a member of the genus *Arctomyoides*.

The morphology of the dentary of UWBM 113209 is broadly similar to that of the dentary of *Protospermophilus oregonensis* (*Downs, 1956*) but the length of the diastema of *Protospermophilus oregonensis* (10.0 mm; *Downs, 1956*) is shorter than that of UWBM 113209 (Table 1). The lower incisor of *Protospermophilus oregonensis* is also slightly deeper (3.7 mm; *Downs, 1956*) than that of UWBM 113209. Finally, UWBM 113209 differs from *Protospermophilus oregonensis* in its lack of striations on the anterior surface of the lower incisor. It thus appears unlikely that the Clarkia rodent represents a specimen of *Protospermophilus oregonensis*. The poor preservation of UWBM 113209 bars a comparison with the diagnostic characters of *Protospermophilus* summarized by *Bryant (1945)* and *Goodwin (2008)*. As such, we cannot exclude the possibility that the Clarkia rodent represents a new large species within the genus *Protospermophilus*.

The cranial morphology of the *Cedromus* material from the Arikareean of Nebraska is poorly known (*Martin, 1973*); so is the skull and dentition of UWBM 113209. As a consequence, it is difficult to assess the similarities between the two similarly-sized squirrels or determine whether or not UWBM 113209 possesses diagnostic characters of the Cedromurinae (*Korth & Emry, 1991*). Nonetheless, prior to the work of *Hayes (2005)*, *Cedromus* was only known from the Orellan and Whitneyan (*Korth & Emry, 1991*) and the assignment of the Arikareean-aged Nebraska material to *Cedromus* is the youngest occurrence of a genus, and subfamily, no less than about 12 million years older than the Clarkia rodent (*Tedford et al., 2004*). The Clarkia rodent therefore likely represents a different taxon than the Arikareean-aged *Cedromus*.

There are four known species of *Protosciurus* (*Goodwin, 2008*). The Clarkia rodent is somewhat larger than *Protosciurus mengi* and *Protosciurus rachelae* (*Hopkins, 2007*; *Korth & Samuels, 2015*). Additionally, the lower incisor of UWBM 113209 is much thicker than the incisors of *Protosciurus mengi* and the skull roof of UWBM 113209 does not display the characteristic supraorbital shelf of *Protosciurus rachelae* (*Korth & Samuels, 2015*). A third species of *Protosciurus, Protosciurus condoni* is much larger than the Clarkia rodent (*Black, 1963*; *Hopkins, 2007*). *Protosciurus condoni* also differs from the Clarkia rodent in its shortened lower diastema (*Black, 1963*). *Protosciurus tecuyensis* is the same size as the Clarkia rodent (*Hopkins, 2007*). The species is known from a single partial lower jaw (*Black, 1963*; *Bryant, 1945*) and, because of the poor morphology of this type and that of the Clarkia specimen, no rigorous comparison between the two squirrels can be undertaken. Despite the morphological differences between the Clarkia rodent and known species of *Protosciurus*, similarities in size and general cranial morphology leave the possibility that UWBM 113209 represents a new species within *Protosciurus*.

The forelimbs of UWBM 113209 are gracile and elongated relative to those of members of the tribe Marmotini including the marmot *Marmota*, the ground squirrel *Spermophilus*, and the prairie dog *Cynomys*; they resemble more closely those of tree squirrels (Tribes Sciurini and Callosciurini; *Bezuidenhout & Evans, 2005*; *Emry & Thorington, 1982*; *Korth & Samuels, 2015*; *Rose & Chinnery, 2004*; *Thorington, Darrow & Betts, 1997*; *Thorington et al., 2005*). Indeed, when accounting for size difference, the forelimb of UWBM 113209 is gracile and more similar to that of the small Clarendonian-aged tree squirrel *Sciurus olsonii* from Nevada, the Arikareean-aged *Protosciurus mengi*, or the modern *Callosciurus prevostii* from southeast Asia than the more robust ground squirrels of the genus *Spermophilus* (*Emry, Korth & Bell, 2005*; *Korth & Samuels, 2015*; *Thorington et al., 2005*). Thus, although the manus of UWBM 113209 is poorly preserved, the visible digits are thin and similar in proportion (although bigger in absolute size) to those of tree squirrels like *S. olsonii*.

## DISCUSSION

The first known tetrapod specimen from the Clarkia *lagerstätte*, UWBM 113209, represents a new occurrence of a rodent from the family Sciuridae. Its discovery not only expands the faunal list of this internationally important locality but increases our understanding of the scope of squirrel diversity during the mid-Miocene. In the absence

of more complete material, and particularly more complete cheek teeth whose morphology can be studied, the Clarkia sciurid cannot currently be assigned to a lower taxonomic level than family. Even so, its large size, proportionately large diastema, shallow dentary, skull shape, and smooth convex lower incisor suggest that UWBM 113209 does not belong to a known sciurid species but might instead represent a new taxon. Only three large-bodied squirrels have been described from the Barstovian: *Arctomyoides, Palaeoarctomyoides*, and *Protospermophilus*, all of which have been interpreted as basal terrestrial squirrels (*Goodwin, 2008*). While the poor preservation of the postcrania of UWBM 113209 precludes a detailed morphometric analysis, its morphology suggests that the Clarkia squirrel is ecologically distinct from previously described mid-Miocene taxa. There is a strong relationship between postcranial morphology and locomotion in extant small mammals including rodents, and the gracile forelimb of UWBM 113209 is a trait correlated with arboreal and scansorial lifestyles (*Chen & Wilson, 2015*; *Samuels & van Valkenburgh, 2008*). The interpretation of the specimen as a tree-dweller is further supported by the paleoenvironment of the Clarkia *lagerstätte*, which preserves a densely forested landscape (*Smiley & Rember, 1985b*). Together, these two lines of evidence suggest that UWBM 113209 is neither a basal terrestrial squirrel nor a ground squirrel (Tribe Marmotini), but possibly a tree squirrel.

If, as its morphology suggests, UWBM 113209 is interpreted as a tree squirrel, it provides insight into a portion of sciurid ecological diversity seldom captured in the fossil record (*Emry, Korth & Bell, 2005*), as most localities of comparable age preserve grassland rather than forest ecosystems (*Strömberg, 2011*). It also illuminates the evolution of tree squirrels during the Miocene. The youngest confirmed occurrence of *Protosciurus* dates back to the Hemingfordian (*Goodwin, 2008*) and the oldest occurrence of *Sciurus* is Clarendonian in age (*Emry, Korth & Bell, 2005*) leaving a hole in the fossil record of tree squirrels during the Barstovian. Recent work in the Mascall Formation of Oregon has uncovered a yet-to-be-described member of the tribe Sciurini that provides evidence for tree squirrels during the Barstovian (*Maguire, Samuels & Schmitz, 2018*). The Clarkia rodent adds to this growing fossil record. The two animals differ in morphology; the Clarkia rodent is much larger (Table 1) than the Mascall sciurin (skull length 48.2 mm; Samuels, 2018, personal communication) but its lower incisor is proportionately not as thick (3.9 mm diameter; J. X. Samuels, 2018, personal communication). Taken together, these two animals suggest the presence of a diverse tree squirrel fauna in the northwestern United States during the early Barstovian that will illuminate the transition from Hemingfordian-aged to Clarendonian-aged tree squirrels.

The Clarkia squirrel is considerably larger than coeval squirrels and the Clarendonian-aged *S. olsonii* (*Emry, Korth & Bell, 2005*); estimates based on lower tooth row length and skull length indicate a body mass of 77.2–85.7 g for *S. olsonii* and of 492 g for UWBM 113209. However, the size of UWBM 113209 is comparable to that of certain species of Arikareean-aged *Protosciurus* (*Goodwin, 2008*; *Hopkins, 2007*; *Korth & Samuels, 2015*). Its estimated mass is also well within the range of modern Sciurini, which range from 81.2 g in *Microsciurus* to 1225 g in *Rheithrosciurus* (*Hayssen, 2008*). The Clarkia squirrel is most similar in estimated mass to the modern *S. alleni, S. aureogaster,*

*S. carolinensis,* and *S. variegatoides.* However, these comparisons do not shed further light on the locomotor ecology of UWBM 113209, as the locomotor ecologies of these species range from largely terrestrial in *S. alleni* to largely arboreal in *S. variegatoides* (*Best, 1995a*, *1995b*).

Recent stratigraphic work at the *lagerstätte* has focused mainly on tephrostratigraphy due to its proximity to several active volcanic centers (*Geraghty, 2017*; *Ladderud et al., 2015*). Because it cannot be definitively identified below the family level, UWBM 113209 is not biostratigraphically informative, but its presence does indicate that conditions favorable to the preservation of small mammals did exist there. Miocene rodents and other small mammals are frequently used to distinguish NALMA subdivisions, including the early Barstovian (*Tedford et al., 2004*). While UWBM 113209 is the only tetrapod recovered from Clarkia to date, future discoveries could allow existing tephrostratigraphic work to be supplemented with biostratigraphic data, further solidifying the age of the *lagerstätte.* Fish, the only vertebrates previously found at Clarkia, have only been reported from the type locality, and even there, they appear only in certain layers (*Smith & Elder, 1985*); UWBM 113209 was recovered from the lowest of these layers. *Smith & Elder (1985)* suggest that, even though some evidence indicates relatively low sedimentation rates in the fish-bearing units, cold temperatures (<10–15 °C) and anoxic conditions favor the preservation of articulated vertebrate specimens in these layers. If this is the case, these layers are the ones that should be targeted in the search for new tetrapod fossils at Clarkia.

## CONCLUSIONS

The specimen described here, UWBM 113209, is a sciurid, and, in all probability, a tree squirrel, making it the first tetrapod of any kind reported from the Clarkia *lagerstätte.* The squirrel represents a significant addition to an already exceptionally preserved mid-Miocene ecosystem. It augments our understanding of a uniquely well-preserved paleo community and the presence of leaves preserved in direct association with the specimen may provide further insight into the nature of species interactions within the Clarkia community. On a broader scale, UWBM 113209 indicates a greater taxonomic and ecological diversity of mid-Miocene Sciuridae than had previously been recognized and provides a unique window onto the paleobiology of infrequently preserved tree squirrels.

## ACKNOWLEDGEMENTS

We would like to thank William Rember for his tireless work on the Clarkia *lagerstätte* and for bringing this specimen to our attention. The Kienbaum family owns the land on which this fossil was found, and we are grateful to them for permitting and facilitating collection on their property. Thanks as well to the staff of the Burke Museum, particularly Meredith Rivin and Gregory Wilson (UWBM), for allowing us to reposit this specimen in the collections in their care. Joshua Samuels reviewed an earlier version of this manuscript and kindly shared unpublished measurements of a squirrel from Oregon. Dale Hanson and editor Andrew Farke also made constructive comments on an earlier version of this article.

### Funding

The authors received no funding for this work.

### Competing Interests

The authors declare that they have no competing interests.

### Author Contributions

- Jonathan J. M. Calede conceived and designed the experiments, performed the experiments, analyzed the data, prepared figures and/or tables, authored or reviewed drafts of the paper, approved the final draft.
- John D. Orcutt conceived and designed the experiments, performed the experiments, prepared figures and/or tables, authored or reviewed drafts of the paper, approved the final draft.
- Winifred A. Kehl contributed reagents/materials/analysis tools, prepared figures and/or tables, authored or reviewed drafts of the paper, approved the final draft.
- Bill D. Richards contributed reagents/materials/analysis tools, prepared figures and/or tables, authored or reviewed drafts of the paper, approved the final draft.

### Data Availability

The specimen described here is reposited in the University of Washington Burke Museum (UWBM), Seattle, Washington, USA.

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
