# Peer review of "The first tetrapod from the mid-Miocene Clarkia lagerstätte (Idaho, USA)"

_PeerJ, doi:10.7717/peerj.4880_

## Round 0.1 · original submission · Minor Revisions

The reviewers have provided a thorough list of suggestions for revision, all of which seem pretty reasonable. Please give all of these consideration as you revise the manuscript, and provide a point-by-point explanation when you resubmit.

In addition to the comments from the reviewers, I would also suggest providing a figure with a close-up of the cranial region. Because the skull features are important for narrowing down the identification of the organism, it would be helpful to have them illustrated with higher resolution.

·

Basic reporting

This article meets the basic reporting guidance, with only minor suggestions/revisions as shown in the attached file.

Experimental design

Very good background for the significance of this fossil specimen, and sufficient discussion to support the research.

Validity of the findings

Sufficient detail to support the information and conclusions. Inconclusive information clearly identified.

·

Basic reporting

This manuscript describes the first tetrapod from the mid-Miocene age Clarkia lagerstatte in Idaho. That site is known for its amazingly well-preserved plant fossils, as well as a variety of insects and fish. The authors describe a specimen, which is accompanied by detailed stratigraphic data and narrow age range, that consists of a carbonaceous film of the body/skeleton and some fragmentary dental material. The material is thoroughly described and the accompanying figures are of high quality.

In the introduction, there are few places where additional references should be cited:
1. Introduction, line 42 – I recommend citing Zachos et al. (2008) as well.

Zachos, J.C., Dickens, G.R., Zeebe, R.E. (2008). An early Cenozoic perspective on greenhouse warming and carbon-cycle dynamics. Nature 451, 279–283.

2. Introduction, line 51 – A recent revision of the Mascall Fauna by Maguire et al. (2018) should be cited here as well.

Maguire, K. C., Samuels, J. X., & Schmitz, M. D. (2018). The fauna and chronostratigraphy of the middle Miocene Mascall type area, John Day Basin. PaleoBios, 35,1–51.

3. Introduction, line 58 – Chaney 1925 is only one of the studies describing the Mascall Flora, the authors should cite Knowlton (1902), Chaney (1959), and Chaney and Axelrod (1959) here as well. Similarly, the Succor Creek Flora (note the name difference from the Sucker Creek Formation) was also more recently analyzed by Fields (1996).

Chaney, R.W. (1959). Miocene floras of the Columbia Plateau. Part I. Composition and interpretation: Carnegie Institute of Washington Contributions to Paleontology, v. 617, p. 1–134.

Chaney, R.W., and Axelrod, D.I. (1959). Miocene floras of the Columbia Plateau. Part II. Systematic considerations: Carnegie Institute of Washington Contributions to Paleontology, v. 617, p. 135–237.

Fields, P.F. (1996). The Succor Creek flora of the Middle Miocene Sucker Formation, southwestern Idaho and eastern Oregon; systematics and paleoecology [Ph.D. dissertation]: East Lansing, Michigan, Michigan State University.

Knowlton, F.H. (1902). Fossil flora of the John Day Basin, Oregon: U.S. Geological Survey Bulletin No. 204.

In the Description, lines 147 to 165 – This text is really not descriptive in nature, rather it represents comparisons to some extant and extinct taxa. As such, the description should be expanded somewhat to describe those postcranial elements and the existing text should be incorporated into the Comparisons section.

In the Comparisons section, the taxonomic assignment and justification for it is well-reasoned, though there are some important omissions that should be addressed:
1. There are currently no comparisons made to Protosciurus, the most common and best-known North American tree squirrel (Sciurini) from the Oligocene and early Miocene. That genus has many records from the Oligocene (Arikareean) and early Miocene (Hemingfordian) (Black 1963, Goodwin 2008, Korth and Samuels 2015). There are even a few questionable records from the mid Miocene (Barstovian), Lezak (1979) and Albright (1996) reported Protosciurus at the Hidalgo Bluff and Trinity River local faunas from the Gulf Coast of Texas.

Three species of Protosciurus are known from skulls: P. condoni, P. mengi, and P. rachelae. While none are complete, they are complete enough to allow comparisons of various dimension to the Clarkia rodent and they also show substantial variation in size. For example, the 3.3 mm lower incisor diameter listed on line 134 is the larger than the lower incisor diameters of 4 specimens of P. mengi from the Arikareean age John Day Formation in Oregon (Korth and Samuels 2015), which range from 2.02 to 2.61 mm. Lower tooth row lengths for those specimens range from 11.49 to 12.72 mm (Korth and Samuels 2015). The lower tooth row length of the type of Protosciurus condoni (UOMNH 5171, described by Black 1963) is 15.56mm. Those sizes bracket the estimate of lower tooth row length provided for the Clarkia rodent in line 174. While none of the Protosciurus skulls are complete, Korth and Samuels (2015) also list occiput to orbit length for P. condoni and P. mengi, which could be compared to the Clarkia specimen.

A partial postcranial skeleton of Protosciurus mengi (UCMP 86368) was also recently described in the article by Korth and Samuels (2015). Like the Clarkia rodent, that specimen displays gracile forelimb elements similar to those of extant tree squirrels.

2. Geomys (Nerterogeomys) can also be ruled out by the fact that it has a flattened chisel-like lower incisor (Gazin 1942, Flynn et al. 2008), while the Clarkia rodent has a convex faced incisor (line 135).

3. There is another mid Miocene tree squirrel (Sciurini) record from the Mascall Fauna of Oregon that was just recently described by Maguire et al. (2018), which is similar in age to the Clarkia record. The subfamily assignment of that specimen was based on aspects of postcranial morphology, in both the humerus and astragalus. Those specimens showed similarity to both extant Sciurus and Protosciurus, and the astragalus (JODA 15793) was most similar to Protosciurus (Maguire et al. 2018). The other specimen (JODA 6725) is a partial skeleton still mostly embedded in a block of matrix. The specimen is exposed enough to allow some measurements, the skull length (48.2 mm) and lower incisor diameter (3.9 mm) are both fairly similar in size to the Clarkia rodent (measurements listed in lines 134 and 139).

There are several places in the text (lines 163, 176, 178, 185) where the terms “Aplodontidae” and “aplodontid” are used. These commonly used in the paleontological literature, but they are technically incorrect. Brandt used “Aplodontidae” in 1855, but In 1896 Thomas corrected the spelling to “Aplodontiidae” to be in accordance with nomenclature codes.

Thomas, O. 1896. On the genera of rodents: An attempt to bring up to date the current arrangement of the order. Proceedings of the Zoological Society of London, 1896: 1012-1028.

Since Thomas, the two names have been used largely interchangeably, but Aplodontiidae is technically correct according to the ICZN code (http://www.nhm.ac.uk/hosted-sites/iczn/code/index.jsp?article=25). 29.3.1 indicates the family name should be the type genus name ("Aplodontia"), minus the suffix ("-a"), with "idae" added at the end. Recent studies like the Wilson and Reeder (2005) Mammal Species of the World and a molecular phylogeny by Piaggio et al. (2013) use “Aplodontiidae” and “aplodontiid”.

Experimental design

No comment

Validity of the findings

The taxonomic assignment and justification for it is well-reasoned, though there are some important omissions in comparisons and discussion that should be addressed. There is a well-known genus of tree squirrels, Protosciurus, that is not mentioned anywhere in the manuscript. With 3 species known from skulls and partial skeletons in the late Oligocene, confirmed records from the early Miocene, and some questionable records from the mid Miocene, it is certainly a relevant taxon to consider. Given the Clarkia specimen's similarity in size and also cranial and postcranial morphology to described specimens of Protosciurus, it suggests that taxon needs to be a key component of the discussion. Since that genus has a number of possible Barstovian occurrences, including one in Oregon, it makes make it an excellent candidate taxon for the specimen from Clarkia.

In the Discussion, line 281 the authors suggest the Clarkia squirrel is a remarkably large one for its age, but I don't think that is actually the case. Early North American squirrels, from the late Eocene and Oligocene, include a number of species larger than the Clarkia specimen, including some very complete Protosciurus specimens from the Arikareean of Oregon mentioned above. That taxon is also known from the Hemingfordian and possibly the Barstovian (Goodwin 2008), and an unidentified tree squirrel (Sciurini) from the contemporaneous Mascall fauna of Oregon is also similar in size to the Clarkia specimen. Sciurus olsonii (Emry et al. 2005) is definitely smaller than the Clarkia specimen and most older tree squirrel records, but it is really the anomaly in terms of tree squirrel size, representing a possible dwarf species like Oligocene species of Miosciurus (Korth and Samuels 2015). Given the fact that the S. olsonii record is more than 3 million year later than Clarkia, the two are certainly not comparable in age.

---

## Round 0.2 · accepted · Accept

Thank you for your close attention to the comments from the reviewers; these improvements to the paper mean that it is now ready for publication (in my opinion).

#